# Sepsis Trajectory Prediction Using Privileged Information and Continuous Physiological Signals

**DOI:** 10.3390/diagnostics14030234

**Published:** 2024-01-23

**Authors:** Olivia P. Alge, Jonathan Gryak, J. Scott VanEpps, Kayvan Najarian

**Affiliations:** 1Department of Computational Medicine and Bioinformatics, University of Michigan, Ann Arbor, MI 48109, USA; 2Department of Computer Science, Queens College, The City University of New York, Flushing, NY 11367, USA; 3Michigan Center for Integrative Research in Critical Care, University of Michigan, Ann Arbor, MI 48109, USA; 4Department of Emergency Medicine, University of Michigan, Ann Arbor, MI 48109, USA; 5Biointerfaces Institute, University of Michigan, Ann Arbor, MI 48109, USA; 6Macromolecular Science and Engineering, University of Michigan, Ann Arbor, MI 48109, USA; 7The Max Harry Weil Institute for Critical Care Research and Innovation, University of Michigan, Ann Arbor, MI 48109, USA; 8Electrical Engineering and Computer Science, University of Michigan, Ann Arbor, MI 48109, USA; 9Michigan Institute for Data Science, University of Michigan, Ann Arbor, MI 48109, USA

**Keywords:** machine learning, privileged information, signal processing

## Abstract

The aim of this research is to apply the learning using privileged information paradigm to sepsis prognosis. We used signal processing of electrocardiogram and electronic health record data to construct support vector machines with and without privileged information to predict an increase in a given patient’s quick-Sequential Organ Failure Assessment score, using a retrospective dataset. We applied this to both a small, critically ill cohort and a broader cohort of patients in the intensive care unit. Within the smaller cohort, privileged information proved helpful in a signal-informed model, and across both cohorts, electrocardiogram data proved to be informative to creating the prediction. Although learning using privileged information did not significantly improve results in this study, it is a paradigm worth studying further in the context of using signal processing for sepsis prognosis.

## 1. Introduction

Sepsis is a condition which presents heterogeneously in different patients, making its diagnosis and prognosis complicated. The Sepsis-3 group defines sepsis as a syndrome that leads to life-threatening organ dysfunction, which is detectable by an increase in two or more Sequential Organ Failure Assessment (SOFA) points. If sepsis progresses to septic shock, a more severe subset, the patient is more likely to experience abnormalities or die [1,2]. It is therefore important to identify patients that are at risk of decompensation to septic shock early, so they can receive necessary care.

One method of early risk identification is the quick-SOFA (qSOFA) method proposed in Sepsis-3, which is a bedside screening tool to identify possible cases of sepsis. The qSOFA score is the sum of three critera: 1 if the Glasgow Coma Scale (GCS) is ≤13, 1 if systolic blood pressure is ≤100 mmHg, 1 if respiratory rate ≥22/min. If the score is ≥2, the patient may be at increased risk for in-hospital mortality or prolonged ICU stay [1,3]. That said, qSOFA is not recommended as a single screening tool for sepsis diagnosis or infection identification [4]. Rather, it is a tool for when data are limited: for example, a patient being newly admitted to hospital, who has not yet had enough labs or vital signs recorded to produce a full SOFA score.

Following the previous study [5], the work presented here focuses on identifying patients at risk to develop poor outcomes related to sepsis. We use a qSOFA score of ≥2 to represent increased risk for poor outcomes. Using a cohort of patients with documented infection, the goal of this work is to predict this move toward increased risk using non-invasive and routinely collected information: namely, electrocardiogram (ECG) and/or electronic health record (EHR) data. Both ECG and EHR data are regularly collected in the intensive care unit (ICU), and so would not add any additional burden to care providers. The distinct advantage offered by ECG, rather than EHR data, is that it is a continuous measure; while different EHR data may be collected at different intervals (every 15 min for vital signs, hourly for fluid output, sporadically for lab tests, etc.), an ECG signal is captured continuously, providing a real-time measure of patient status.

This work also builds upon the continuous nature of ECG by including privileged information (PI), which, in a machine learning context, means data that are available at the training stage but not at the validation or testing stages. Because the dataset used for model training is retrospective, we can let our model view future events for the training cases in order to improve predictions on the test set. Learning using PI is described further in Section 2.1.2. While learning using privileged information (LUPI) has existed for many years, and has been used in other medical applications [6,7], it has not yet been used in the context of signal processing for sepsis prognosis. This paper offers an insight into a potential application of LUPI for sepsis prognosis.

## 2. Methods

### 2.1. Machine Learning

The basic model used for machine learning was support vector machine (SVM) [8]. SVM was selected for two main reasons: first, its transparency and interpretability, and second, the availability of the SVM+ extension of SVM, to serve as a comparison method with privileged information. SVM is well understood and widely tested, making it an ideal baseline; SVM+ is a relatively straightforward expansion of SVM to the privileged space. Therefore, using these models as a baseline, we can later test if more complex models of learning using privileged information should be pursued for this particular sepsis prognosis model.

For all models trained, we used a Gaussian kernel with the sequential minimal optimization [9] solver. A grid search selected a box constraint and kernel scale that resulted in the greatest area under the receiver operating characteristic curve (AUROC) value in the validation set. The process of model training was repeated 100 times. In each iteration, the dataset was divided patient-wise into distinct training, validation, and test sets, such that no patient in one set (training, test, or validation) could appear in another. The test set was withheld from model training. The mean and standard deviation of F1 score, sensitivity, specificity, AUROC, and area under the precision–recall curve (AUPRC) over 100 iterations were recorded, and these are reported in Section 3.

#### 2.1.1. Support Vector Machine

The model used to benchmark performance is the SVM with a Gaussian kernel. To learn the decision rule y=f(x), it maps vectors of x∈X into vectors z∈Z and constructs the optimal separating hyperplane between the two classes. The optimal separating hyperplane between the two classes is constructed by learning the decision rule f(z)=wz+b, where wandb are parameters of the hyperplane (weight and bias, respectively), and SVM’s objective function is: (1)$$\min_{w,b,\xi }\frac{1}{2}{∥w∥}^{2}+C\sum_{i}^{n}\xi_{i}$$
with the constraints
$$\begin{array}{cc} y_{i}\left(w\cdot z_{i}+b\right) & \geq 1-\xi_{i},\;i=1,\ldots ,n\\ \xi & \geq 0,\;C>0 \end{array}$$
where $\left(x_{i},y_{i}\right)$ are a sample’s input and label pair, $\xi_{i}$ functions as a slack variable and *C* is the penalty parameter [8]. These allow for soft-margin decision boundaries when classes are not linearly separable.

#### 2.1.2. Learning Using Privileged Information

In this paper, we also use an expanded version of the SVM algorithm, SVM+. The implementation of SVM+ used in this paper comes from [10], which was a modified version of the SVM+ algorithm developed in [11], which in turn was an extension of SVM [8].

As defined by Vapnik and Vashist [11], LUPI is a paradigm where, in the training stage, the teacher presents both training example *x* as well as additional information $x^{*}$ to the learner:$$x_{1},\ldots ,x_{n}\in X\;\mathrm{and}\;x_{1}^{*},\ldots ,x_{n}^{*}\in X^{*},$$
where *n* is the number of samples in the training set, and *X* and $X^{*}$ are different spaces. Privileged information is not included in the test or validation sets. Vapnik and Vashist go on to define the paradigm as: when given a set of triplets $\left(x_{1},x_{1}^{*},y_{1}\right),\ldots ,\left(x_{n},x_{n}^{*},y_{n}\right)$, where y∈−1,1 is the classification created according to unknown probability measure $P(x,x^{*},y)$, find the function y=f(x,α∗),α∈Λ that guarantees the smallest probability of incorrect classification.

Building on how SVM maps x∈X to z∈Z, SVM+ maps privileged information $x^{*}\in X^{*}$ to $z^{*}\in Z^{*}$. The objective function of SVM+ is: (2)$$\min_{w^{*},b^{*},w,b}\frac{1}{2}\left({∥w∥}^{2}+\gamma {∥w^{*}∥}^{2}\right)+C\sum_{i}^{n}\xi \left(w^{*},b^{*},z_{i}^{*}\right)$$
such that
$$\begin{array}{cc} y_{i}\left(w\cdot z_{i}+b\right) & \geq 1-\xi \left(w^{*},b^{*},z_{i}^{*}\right),\\ \xi \left(w^{*},b^{*},z_{i}^{*}\right) & \geq 0,\\ \gamma & >0 \end{array}$$
where $\xi \left(w^{*},b^{*},z_{i}^{*}\right)=w^{*}\cdot z_{i}^{*}+b^{*}$ is the slack function for the privileged space, replacing the slack variables $\xi_{i}$, and γ is a hyperparameter. From this, the hyperplane of SVM+ can be tuned by PI, as privileged training samples $x_{i}^{*}$ can be used to regularize the loss from training samples $x_{i}$.

Li et al. expanded upon SVM+’s implementation to create an efficient sequential minimal optimization algorithm to solve it [10]. Once the feature vectors are augmented into nonlinear space (creating $z_{i}\leftarrow {\left[z_{i}^{⊤},1\right]}^{⊤}$ and $w\leftarrow {\left[w^{⊤},b\right]}^{⊤}$ in the regular space and $z_{i}^{*}\leftarrow {\left[z_{i}^{*⊤},1\right]}^{⊤}$ and $w^{*}\leftarrow {\left[w^{*⊤},b^{*}\right]}^{⊤}$ in the privileged space), they represent the decision function as f(x)=w·z and use squared hinge loss to create the following formulation:(3)$$\min_{w^{*},w,\rho }\frac{1}{2}\left({∥w∥}^{2}+\gamma {∥w^{*}∥}^{2}\right)+\frac{1}{2}C\sum_{i}^{n}{\left(w^{*}\cdot z_{i}^{*}\right)}^{2}-\rho$$
where ρ is a value such that $y_{i}\left(w\cdot z_{i}\right)\geq \rho -\left(w^{*}\cdot z_{i}^{*}\right)$, which they proceed to solve using its dual formation.

The dual form of Equation (Equation 3) is based on its Lagrangian,
(4)$$L=\frac{1}{2}\left({∥w∥}^{2}+\gamma {∥w*∥}^{2}\right)+\frac{1}{2}C\sum_{i}^{n}{\left(w^{*}\cdot z_{i}^{*}\right)}^{2}-\rho -\sum_{i}^{n}\alpha_{i}\left(y_{i}\left(w\cdot z_{i}\right)-\rho +w^{*}\cdot z_{i}^{*}\right)$$
where $\alpha =[\alpha_{1},\ldots \alpha_{n}]$. When the derivatives of Equation (Equation 4) are set with respect to the primal variables $w,w^{*},\rho$ to zeros, the following are obtained:$$\begin{array}{cc} \mathrm{the}\mathrm{constraint}\\ \alpha^{⊤}1=1\\ \mathrm{the}\mathrm{Karush}–\mathrm{Kuhn}–\mathrm{Tucker}\mathrm{conditions}\\ w= & \sum_{i}^{n}\alpha_{i}y_{i}z_{i}\\ \mathrm{and}\\ w^{*}= & \sum_{i}^{n}\alpha_{i}{\left(\gamma I+CPP^{⊤}\right)}^{-1}z_{i}. \end{array}$$

When the previous two equations for *w* and $w^{*}$ are substituted into Equation (Equation 4), this yields
(5)$$\min_{\alpha }\frac{1}{2}\alpha^{⊤}\left(H+G\right)\alpha$$
where
$$\begin{array}{cc} \alpha & \geq 0\\ 1^{⊤}\alpha & =1\\ G & =P^{⊤}{\left(\gamma I+CPP^{⊤}\right)}^{-1}P\\ P & =[z_{i}^{*},\ldots z_{n}^{*}]\\ H & =K\circ (yy^{⊤}) \end{array}$$
and **K** is the kernel matrix of augmented features in the regular space [10]. The value of α is obtained by using the dual form of one-class SVM, where Q is the kernel matrix, ν is a pre-defined variable, and *n* is the number of training samples:(6)$$\;\;\min_{\alpha }\frac{1}{2}\alpha^{⊤}Q\alpha$$
(7)   suchthat
(8)$$\;\;\;1^{⊤}\alpha =\nu n$$
(9)and 0≤α≤1.

Learning using PI in a medical context builds upon previous work [6,7,10]. The features that we included as PI are described in Section 2.3.3.

### 2.2. Dataset

The data used in this study were obtained from a retrospective dataset created by the University of Michigan, and this data collection was approved by the institutional review board (IRB) of University of Michigan. Because the nature of the study (creating a biobank consisting of previously collected and de-identified data) was retrospective and did not directly involve human subjects, informed consent was waived. Demographics information is presented in Appendix A Table A1.

The dataset from the biobank consisted of 1803 unique individuals age ≥18 years with 3516 unique encounters between 2013 and 2018 at Michigan Medicine. Individuals reported their own sex and race/ethnicity from categories defined by Michigan Medicine. The detailed inclusion/exclusion criteria for the dataset were provided in [5], but briefly: inclusion criteria selected for inpatient encounters with ECG lead II waveforms at least 15 min in length and ICD 9/10 codes for pneumonia, cellulitis, or urinary tract infection (UTI), excluding UTIs associated with catheters. Exclusion criteria included positive HIV status, solid organ or bone marrow transplant, and ongoing chemotherapy. The criteria selected were defined as to create a dataset that could capture patients with an infection at risk to decompensate to septic shock rather than select for a sepsis diagnosis outright. The full list of ICD codes used to construct the full dataset are presented in Appendix D.

We used increase in qSOFA score to assign positive and negative classes. Given an individual who met one of the criteria for qSOFA, the SVM would predict whether their qSOFA score would increase to 2 or 3 after a prediction gap of six hours. An increase in qSOFA was considered the positive outcome in a learning context, and the negative outcome was qSOFA <2 after the prediction gap.

We created two cohorts from the retrospective dataset. The first is modeled after the dataset from [5], consisting of critically ill patients in the ICU. The second is a broader, more heterogeneous dataset in the ICU. Because sepsis does present differently among patient groups, we were curious to see if incorporating privileged information would be more helpful in the broader context (cohort 2) or in a more specific, defined patient group (cohort 1).

#### 2.2.1. Cohort 1

To create this cohort from the full dataset, we selected for individuals with EHR, ECG, and arterial line data available 10 min before and up to $t_{0}$ as well as 10 min before and up to $t_{6}$. In this study, EHR data included labs, medications, hourly fluid output, and vital signs. Upon collecting 10-min signals for feature extraction, signals determined to be 50% or more noise were discarded.

With these conditions in place, the final dataset consisted of 106 instances of 105 patients with 59 positive cases and 47 negative cases. Due to the small size of the cohort, we opted to use repeated train/test splits rather than 3-fold cross-validation as in [5]. The train/test split was 80/20 with a further 20% of the train set being reserved as a validation set for the grid search.

#### 2.2.2. Cohort 2

Due to the small size of cohort 1, we created cohort 2 with more relaxed criteria. Namely, we only selected for individuals with EHR and ECG available in both the regular and privileged space and omitted the requirement for arterial line data. This cohort 2 consisted of 453 instances of 434 unique individuals with 144 positive cases and 309 negative cases. We used a similar train/test split as in cohort 1. We created this second, larger cohort for two reasons: (1) to see if ECG- and EHR-related results were consistent across both cohorts, and (2) as an arterial line is typically only used for critically ill patients [12], we wanted to validate our findings on a greater variety of patients with different statuses.

### 2.3. Signal Processing

For every data sample, we collected the 10 min of ECG signal occurring directly before the prediction gap for processing. This 10-min signal was divided into 2 5-min windows. This constitutes the signal collected in the regular space. For signals collected in the privileged space, we used the 10 min of ECG signal directly at the end of the prediction gap, that is, a 10-min period that ends at the event time, $t_{6}$.

#### 2.3.1. Electrocardiogram Preprocessing

ECG data consisted of four leads sampled at 240 Hz. We used lead II of the ECG for the analysis and filtered it with a second-order Butterworth bandpass filter with the cutoff frequencies 0.5 and 40 Hz to remove noise and artifacts, following previous work [13,14]. When 10-min periods of ECG signal were collected, these 10 min were divided into two 5-min windows.

#### 2.3.2. Feature Extraction in the Regular Space

In this paper, the regular space was information available at or before time $t_{0}$, which was when qSOFA was recorded as being equal to 1. Anything occurring after $t_{0}$ was considered the privileged space. An illustration is provided in Figure 1 to show where the regular and privileged space for this particular experiment appeared on a timeline.

We calculated peak-based and statistical features from the Taut String (TS) approximation [15] of each window of the 10-min signal captured six hours before the increase of qSOFA. These TS features have been used in prior work within the healthcare context [5,13,14,16].

Given a discrete signal $f=(f_{0},f_{1},\ldots ,f_{n})$ and a fixed value ϵ>0, the TS estimate of *f* is the unique function *g* such that
$${∥f-g∥}_{\infty }=\max_{i}\left\{\left|f_{i}-g_{i}\right|\right\}\leq \epsilon$$
and
$${∥D\left(g\right)∥}_{2}=\sqrt{\sum_{i=1}^{n-1}{\left(g_{i+1}-g_{i}\right)}^{2}}$$
is minimal, with *D* being the difference operator. This produces a piecewise linear estimation that appears like a string being pulled tightly between the peaks and valleys of the original input signal. When the TS estimate is subtracted from the original input signal, this produces a “noise” estimation.

TS estimation was applied to each window of the filtered ECG signal using five values of the parameter ϵ: 0.0100, 0.1575, 0.3050, 0.4525, and 0.6000, which were chosen from previous work [13,14]. Six features were computed from each TS estimate of a 5-min window and value of ϵ. These features included the following: number of line segments, number of inflection segments, total variation of noise, total variation of denoised signal, power of denoised signal, and power of noise, resulting in a tensor of size 2×5×6 for each signal, where the modes of the tensor were window, ϵ, feature.

In addition to signal features, EHR data features were also collected from both the 10 min before $t_{0}$ as well as four additional lookback periods at $t_{-4},t_{-8},t_{-12},\mathrm{and}\;t_{-16}$. These were based on those used in [5] and included the following: ordinal encoding of lab values (creatinine, glucose, hematocrit, hemoglobin, international normalized ratio, lactate, platelet count, potassium, sodium, white blood cell count) that ranged from 1 to 4 in increasing severity (with 0 indicating nothing logged); ordinal encoding of cardiovascular infusions (dobutamine, dopamine, epinephrine, isoproterenol, milrinone, norepinephrine, vasopressin) that ranged from 1 to 3 in increasing severity, and readings of vital signs (heart rate, blood pressure, temperature, SpO_2_) and hourly urine output. The ordinal encoding of cardiovascular infusions is detailed in Appendix B Table A2, and the ordinal encoding of labs is shown in Appendix C Table A3. The full list of features extracted from the regular space is shown in Table 1.

#### 2.3.3. Feature Extraction in the Privileged Space

To generate features in the privileged space, ten minutes of ECG signal were extracted starting from ten minutes before the event of interest up to the event ($t_{6}-10$ min to $t_{6}$). The signal underwent the same Butterworth bandpass filter as the regular space ECG data. Two different sets of features were computed from this period of time in the ECG signal. The first set contained a statistical summary features: mean, median, variance, kurtosis, skewness, Shannon entropy, and the mean absolute value of the fast Fourier transform (FFT). These were adapted from a set of features computed in [13]. This first set of features is referred to as the set of SF-ECG privileged features with “SF” standing for “statistical features”.

To create the second set of features, we applied TS to this 10-min signal from the privileged space. Using the same ϵ values as from Section 2.3.2, we computed the number of line segments, number of inflection segments, total variation of noise, total variation of denoised signal, power of denoised signal, and power of noise over the 10-min segment. This second set of features is called the TS-ECG privileged features with “TS” standing for “Taut String”.

Lastly, one more set of features was computed from the privileged space: EHR data features. These features were the same as those computed in the regular space (labs, cardiovascular infusions, fluid output, vital signs), but they did not include the four sets of lookback features; instead, these were only collected from the ten minutes of privileged space. The full list of features extracted from the privileged space is shown in Table 2.

## 3. Results

The tables included here show the results of SVM models and SVM+ models trained with different types of privileged information. In each table, the PI Type“none” indicates a basic SVM model with a Gaussian kernel. All other PI types use SVM+ to incorporate the privileged information.

For models trained on cohort 1 with Taut String ECG data, shown in Table 3, using SVM+ with additional Taut String ECG privileged information increased the average AUROC by 0.03 and average AUPRC by 0.02, with standard deviation remaining similar, compared to the base SVM model. Cohort 2 does not show this increase with PI, but rather, it yields the highest AUROC and F1 score when no PI is added. Although cohort 2’s average F1 score, AUROC and AUPRC are lower than cohort 1’s, the standard deviation for each is smaller, as shown in Table 4.

Models trained on both TS and EHR are shown in Table 5 for cohort 1 and Table 6 for cohort 2. Neither model shows improvement upon adding PI. Cohort 1’s results show a greater F1 score and AUPRC compared to cohort 2’s results, but cohort 2 has an increased AUROC with smaller standard deviations across all values.

For models trained on EHR data in cohort 1, as shown in Table 7, adding privileged EHR data increased mean F1 score, AUROC, and AUPRC by 0.03 with standard deviation decreasing in all cases. In cohort 2, adding PI did not improve performance. However, AUROC is higher and with a smaller standard deviation compared to cohort 1, as shown in Table 8.

## 4. Discussion

For the two cohorts in the previous sections, we found differing effects of adding PI to an SVM model. In cohort 1, the smaller cohort selected for patients more likely to be critically ill, adding taut string privileged information was slightly beneficial when ECG alone was being used as the regular space (AUROC 0.68 ± 0.12 compared to 0.65 ± 0.13). In cohort 2, the larger and broader cohort, PI was not as informative to the models in any of the presented scenarios.

For cohort 1, the TS-ECG SVM+ model with ECG as the regular space outperformed EHR in the regular space and ECG and EHR in the regular space across F1 score, AUROC, and AUPRC. Cohort 2 had more positive influence from EHR data, where the models including both ECG and EHR data in the regular space outperformed any variation of ECG or EHR data alone in the regular space regardless of adding PI. In both cohorts, ECG information is strongly contributing to the model.

It is possible that the EHR data are more informative in the broader cohort as the patients are more diverse; critically ill patients would be receiving similar antibiotic, vasopressor, and other therapies, and therefore, EHR data would be similar across all patients, whereas a broader patient cohort may have different treatments being given to them, making EHR data more distinctive between the more and less severe cases.

Cohort 1 was initially selected with the goal of also including arterial line features as both regular space and privileged information features; however, neither of these features significantly improved performance compared to the models only trained on ECG data.

It is also noted that in addition to the dataset being somewhat small, when constraints based on signal availability are created, the dataset also loses racial and ethnic diversity, with the vast majority of the cohort being made of white individuals, although the distribution of sex was roughly equal. Studies of sepsis prognosis using LUPI should be replicated on both larger and more diverse cohorts outside of this one particular hospital to ensure that results are generalizable to a greater patient population.

The slight improvement found in cohort 1 with PI shows that incorporating privileged information into a sepsis prognosis clinical decision support system has potential; however, this particular approach of using qSOFA as a proxy variable for risk to decompensate may be lacking. Future trials could investigate different lengths of signal or windowing parameters, or considering different lookback periods for historical ECG collection, such as extending to 24 h or earlier in a patient’s EHR. Additionally, different designs of PI collection can be explored. For example, Sabeti et al. have used LUPI where PI is only available for certain samples, using a “learning using partially available privileged information” paradigm [6,7]. Lastly, different outcome variables, such as start of mechanical ventilation, vasopressor administration, change of antibiotic dose, or others which are clinically relevant, could be studied with an LUPI approach.

We do not want to dismiss an LUPI approach to sepsis prognosis outright, but rather, focus future work on fine tuning a signal- and/or EHR data-informed clinical decision support system for sepsis prognosis. LUPI presents the opportunity to fine tune a model using historical patient data when limited data are available in the current moment (i.e., the training set, or a patient currently in ICU with unknown trajectory), and as such, is a potentially powerful tool. Future study is needed to determine the most practical application of LUPI in the ICU for sepsis prognosis.

## Figures and Tables

**Figure 1 diagnostics-14-00234-f001:**
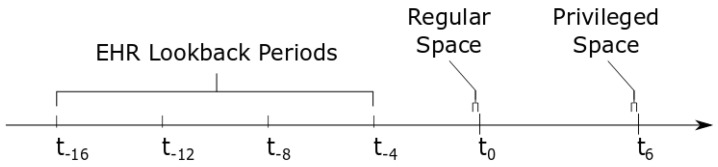
Illustration of timeline. Here, $t_{0}$ is the point where qSOFA is 1, and $t_{6}$ is six hours later, where qSOFA either increases to 2 or 3 (positive) or not (negative). The times $t_{-4}$ to $t_{-16}$ are lookback periods included in the EHR data in the regular space. The brackets at time $t_{0}$ show the ECG signal collected in the regular space, *x*, and the brackets at time $t_{6}$ show the ECG signal collected in the privileged space, $x^{*}$.

**Table 1 diagnostics-14-00234-t001:** Features computed in the regular space.

Group	Feature	Time(s) Collected	Type
Taut String ECG Features	Number of Line Segments,	$t_{0}$	Numerical
	Number of Inflection Segments,		
	Total Variation of Noise,		
	Total Variation of Denoised Signal,		
	Power of Noise,		
	Power of Denoised Signal		
EHR Vital Signs	Temperature,	$t_{-16}$, $t_{-12}$, $t_{-8}$, $t_{-4}$, $t_{0}$	Numerical
	SpO_2_,		
	Heart Rate,		
	Mean Arterial Pressure,		
	Respiratory Rate		
EHR Fluid Output	Urine Output	$t_{-16}$, $t_{-12}$, $t_{-8}$, $t_{-4}$, $t_{0}$	Numerical
EHR Lab Values	Creatinine,	$t_{-16}$, $t_{-12}$, $t_{-8}$, $t_{-4}$, $t_{0}$	Ordinal
	Glucose,		
	Hematocrit,		
	Hemoglobin,		
	INR *,		
	Lactate,		
	Platelet Count,		
	Potassium,		
	Sodium,		
	WBC **		
EHR CVIs ***	Dobutamine,	$t_{-16}$, $t_{-12}$, $t_{-8}$, $t_{-4}$, $t_{0}$	Ordinal
	Dopamine,		
	Epinephrine,		
	Isoproterenol,		
	Milrinone,		
	Norepinephrine,		
	Vasopressin		

* INR = International Normalized Ratio, ** WBC = White Blood Cell Count, *** CVIs = Cardiovascular Infusions.

**Table 2 diagnostics-14-00234-t002:** Features computed in the privileged space.

Group	Feature	Time(s) Collected	Type
Taut String ECG Features	Number of Line Segments,	$t_{6}$	Numerical
	Number of Inflection Segments,		
	Total Variation of Noise,		
	Total Variation of Denoised Signal,		
	Power of Noise,		
	Power of Denoised Signal		
Statistical ECG Features	Mean,	$t_{6}$	Numerical
	Median,		
	Variance,		
	Kurtosis,		
	Skewness,		
	Shannon Entropy,		
	Absolute Value of FFT		
EHR Vital Signs	Temperature,	$t_{6}$	Numerical
	SpO_2_,		
	Heart Rate,		
	Mean Arterial Pressure,		
	Respiratory Rate		
EHR Fluid Output	Urine Output	$t_{6}$	Numerical
EHR Lab Values	Creatinine,	$t_{6}$	Ordinal
	Glucose,		
	Hematocrit,		
	Hemoglobin,		
	INR *,		
	Lactate,		
	Platelet Count,		
	Potassium,		
	Sodium,		
	WBC **		
EHR CVIs ***	Dobutamine,	$t_{6}$	Ordinal
	Dopamine,		
	Epinephrine,		
	Isoproterenol,		
	Milrinone,		
	Norepinephrine,		
	Vasopressin		

* INR = International Normalized Ratio, ** WBC = White Blood Cell Count, *** CVIs = Cardiovascular Infusions.

**Table 3 diagnostics-14-00234-t003:** Taut string ECG in the regular space with different types of privileged information available in cohort 1.

PI Type	F1 Score	Sensitivity	Specificity	AUROC	AUPRC
None	0.71 (0.10)	0.71 (0.16)	0.65 (0.17)	0.65 (0.13)	0.66 (0.10)
TS-ECG	0.70 (0.11)	0.69 (0.16)	0.69 (0.14)	0.68 (0.12)	0.68 (0.10)
SF-ECG	0.70 (0.10)	0.68 (0.15)	0.68 (0.15)	0.65 (0.12)	0.67 (0.12)
EHR	0.70 (0.12)	0.70 (0.18)	0.66 (0.15)	0.65 (0.13)	0.66 (0.11)

**Table 4 diagnostics-14-00234-t004:** Taut String ECG in the regular space with different types of privileged information available in cohort 2.

PI Type	F1 Score	Sensitivity	Specificity	AUROC	AUPRC
None	0.51 (0.06)	0.62 (0.10)	0.63 (0.10)	0.62 (0.07)	0.42 (0.07)
TS-ECG	0.48 (0.06)	0.60 (0.11)	0.59 (0.10)	0.58 (0.07)	0.39 (0.07)
SF-ECG	0.50 (0.06)	0.62 (0.10)	0.58 (0.10)	0.59 (0.07)	0.39 (0.06)
EHR	0.51 (0.07)	0.64 (0.10)	0.59 (0.09)	0.61 (0.08)	0.40 (0.08)

**Table 5 diagnostics-14-00234-t005:** Results of ECG and EHR in the regular space for cohort 1.

PI Type	F1 Score	Sensitivity	Specificity	AUROC	AUPRC
None	0.69 (0.09)	0.66 (0.13)	0.71 (0.14)	0.65 (0.11)	0.66 (0.10)
TS-ECG	0.69 (0.10)	0.67 (0.15)	0.70 (0.14)	0.64 (0.12)	0.65 (0.09)
SF-ECG	0.68 (0.10)	0.67 (0.15)	0.67 (0.14)	0.62 (0.12)	0.65 (0.10)
EHR	0.68 (0.11)	0.66 (0.16)	0.69 (0.16)	0.63 (0.13)	0.65 (0.10)

**Table 6 diagnostics-14-00234-t006:** Results of ECG and EHR in the regular space for cohort 2.

PI Type	F1 Score	Sensitivity	Specificity	AUROC	AUPRC
None	0.60 (0.06)	0.70 (0.09)	0.69 (0.09)	0.72 (0.05)	0.54 (0.08)
TS-ECG	0.58 (0.05)	0.70 (0.09)	0.67 (0.08)	0.70 (0.06)	0.50 (0.08)
SF-ECG	0.58 (0.05)	0.69 (0.08)	0.68 (0.08)	0.71 (0.05)	0.51 (0.07)
EHR	0.59 (0.05)	0.70 (0.09)	0.69 (0.08)	0.72 (0.05)	0.53 (0.06)

**Table 7 diagnostics-14-00234-t007:** Results of EHR in the regular space for cohort 1.

PI Type	F1 Score	Sensitivity	Specificity	AUROC	AUPRC
None	0.59 (0.15)	0.59 (0.21)	0.61 (0.18)	0.51 (0.13)	0.55 (0.10)
TS-ECG	0.59 (0.13)	0.59 (0.19)	0.58 (0.17)	0.49 (0.12)	0.54 (0.09)
SF-ECG	0.62 (0.11)	0.61 (0.17)	0.60 (0.17)	0.51 (0.12)	0.56 (0.09)
EHR	0.62 (0.12)	0.59 (0.17)	0.64 (0.17)	0.54 (0.12)	0.58 (0.09)

**Table 8 diagnostics-14-00234-t008:** Results of EHR in the regular space for cohort 2.

PI Type	F1 Score	Sensitivity	Specificity	AUROC	AUPRC
None	0.59 (0.06)	0.68 (0.09)	0.71 (0.10)	0.71 (0.06)	0.55 (0.09)
TS-ECG	0.55 (0.06)	0.67 (0.10)	0.65 (0.09)	0.67 (0.07)	0.47 (0.07)
SF-ECG	0.56 (0.06)	0.68 (0.09)	0.66 (0.07)	0.68 (0.06)	0.49 (0.08)
EHR	0.57 (0.06)	0.66 (0.09)	0.68 (0.10)	0.68 (0.06)	0.51 (0.08)

## Data Availability

The data that support the findings of this study belong to the University of Michigan and cannot be publicly distributed due to reasons of patient privacy. Data are located in controlled access data storage at the University of Michigan. Restrictions apply to the availability of these data and may be made available with the permission of the University. Any requests regarding access to data from this study may be sent to Drew Bennett (andbenne@umich.edu) of the Universiy of Michigan Innovation Partnerships.

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
