# Peer review of "Sepsis Trajectory Prediction Using Privileged Information and Continuous Physiological Signals"

_diagnostics, 2024, doi:10.3390/diagnostics14030234_

Round 1
Reviewer 1 Report
Comments and Suggestions for Authors
The authors present a ML framework to leverage privilege information while training a model to predict a small vs. big increase in qsofa score.
I have listed many questions and provided comments below. Few of my main concerns are: one, the paper is lacking the motivation for why privilege information would benefit sepsis prognosis more than any other type of information; two, the methods/protocols are not scientifically sound (see questions/comments below); three, there is absolutely no investigation on why we are seeing increase / decrease in performance in Tables 1 through 6; four, all numbers are in a very similar ball park that (I hypothesize) the effects will wash out if you attempt to evaluate on an external dataset or on a larger dataset.
While this framework is very interesting and has the potential to improve model performance across a whole set of problems the generalizability of this framework is questionable. Can this framework be applied to other ML models (e.g., gradient boosted trees) which are very popular with ICU clinicians?
Questions
- Motivate upfront why you chose SVM. It appears you chose SVM since Li et al., made an efficient implementation of SVM+ to take PI into account
- Not sure why you consider only qsofa scores that go from 1 to 2 (negative example) or 1 to 3 (positive example). What about qsofa scores that go from 0 to 2 (negative) or 0 to 3 (positive). Please clarify
- No statistics on prevalence of qsofa score. But from cohort 2 I infer that few patients had multiple data samples while most patients had just one data sample. How often is qsofa available? I realize SBP and RR are measured frequently so qsofa should be restricted by the presence of GCS score. Can GCS score be forward filled and hence qsofa computed multiple times per patient? This would increase the number of data samples and hence it is crucial to discuss. Please clarify
- What was the prevalence of ECG in the two five minute windows in the regular space and privilege space? Were there any windows that were dropped? How did you combine the two five minute windows? It appears you applied Taut string analysis to each five minute window
- In cohort 1 (and to some extent cohort 2) the patient count is small that a leave-one-patient-out might be a better cross validation protocol than the random train-validation-test split you performed. Why? More data becomes available to train each model, so there is less variance amongst train datasets and hence this leads to simpler models that do not overfit to noise inherent in small datasets.
- What are the bin boundaries for labs when binning them into 1-4, similarly for vasopressors going from 1-3? Please clarify
- Labs (e.g., arterial blood gases) in ICU are typically measured every 24 hours. So it might make more sense to look back 24 hours for EHR data
- What is the lookback window for EHR for privileged feature extraction? Looks like you eating into the six hour gap window. Is this desirable?
- Is there absolutely no difference between SVM and SVM+ implementation? Have you done a head-to-head comparison? If not then it makes sense to use SVM+ throughout your paper and use dummy features (like all ones or all zeros) in place of “PI-Type == None”. Right now, the difference in performance between the first row and rows 2-4 across all tables, can be attributed to the differences between SVM and SVM+ (equations 1 vs. 3)
- I think readers would really benefit by having a feature count per analysis. I highly suspect the number of features would exceed the number of data samples in Experiment 2 (Table 3 and 4). Please add
- For results in Tables 3 and 4 are you using the EHR data from all four windows (-4, …, -16)? My hunch is that most labs would remain identical across the four time windows and hence are simply redundant. I see you are taking a l2 norm on the weights, but you could reduce complexity by performing feature selection upfront
- Why are you not combining the TS-ECG, SF-ECG and EHR features?
- While mere presence of ABP might indicate critically ill patients there are other approaches to detect critically ill patients (e.g., APACHE score)
Easy to read and follow.
Minor edits
- Fix “can be which available …”
- Fix “look view future …”
- Fix “Basic model that used …”
- Replace “For every sample, we …” with “For every data sample, we …”
- Fix “Taut string being pulled taught”
Author Response
- SVM+ is a standard, relatively straightforward, and well-tested method of using privileged information. By showing that results are satisfactory with SVM+, we can further branch out to other more complex implementations of privileged information. The first paragraph of the methods section has been updated to reflect this.
- The negative group is qSOFA scores that remain as "not severe", i.e., remain at score 1 or 0. Scores that increase from 1 to 2 or 1 to 3 are positive. We start at qSOFA of 1, as this is the case that could quickly turn from "not severe" to "severe". In this cohort, there were very few instances of going from qSOFA score of 0 to 2 or 3, but this can be considered in future iterations.
- Yes, we do forward-fill GCS. There were not many cases of qSOFA score rising to >= 2 and later falling and rising once again, so there was usually one instance of qSOFA increasing per patient. The lead author no longer has access to the dataset and so cannot provide the numbers of how often qSOFA is available.
- Individuals that did not have 10 continuous minutes of ECG data in both the regular and privileged space were dropped before performing Taut String analysis, so all patients in the final dataset had ECG data. Yes, Taut String was applied to each five-minute window. This built off previous work in (Alge et al, reference 5) which also used five-minute windows for analysis.
- Authors will consider leave-one-out cross validation in future work with this dataset.
- We originally did not include this information so as not to make it redundant with reference 5. Tables showing the reference ranges for labs and vasopressors have now been added to the Appendices.
- The authors note that labs in the ICU can be collected more frequently than 24 hours, with arterial blood gases drawn every two hours depending on the stability of the patient. That being said, in future work we can consider different lookback periods.
- No, we are not going into the six-hour gap window. The EHR privileged features do not include lookback windows. See lines 223-224, "did not include the four sets of lookback features".
- The basic structure of SVM and SVM+ are very similar, thus why we use the comparison. While a future analysis could compare using SVM with both the privileged and regular space features as input, that defeats the purpose of the experiment, which is to simulate the hospital setting where the privileged information would not be available for newly admitted patients. On the other hand, SVM is much faster to create and test than SVM+, and so there would not be a use case for an SVM+ model trained with dummy variables in an actual hospital setting. If SVM+ is not useful with potentially informative variables, then the next logical step would be to use the faster, well-tested and reliable standard SVM.
- We have now added tables (Table 1, page 8, Table 2, page 9) that list all of the features extracted from the regular and privileged spaces.
- The authors agree that lookback periods do tend to show that labs are similar, however, in the case where no labs are performed (giving a value of 0) and then labs are later performed (updated value), this does show a change. Additionally, in patients with less stability, labs are being performed more frequently, and thus, show more variability.
- Our team did test combinations of ECG and EHR features together, and did not find them to perform any better than any of the features individually in the privileged space. As such, these results were omitted. Similarly, tests including ABP as a feature in either privileged or regular space did not show any improvement compared to ECG or EHR data, and those results were also omitted.
- The authors note that the APACHE score is useful as a research tool, but in practice, requires too much time and too many variables to be helpful as a clinical decision making tool in the ICU. The selection of ABP as a deciding factor was also to serve as a direct follow-up to our previous work which used ABP as inclusion criteria (Alge et al, reference 5).
- Grammatical errors have been fixed.
Reviewer 2 Report
Comments and Suggestions for Authors
The manuscript offers unique insight on rapid assessment of sepsis and explores the role of ECG and EHR information in a SVM structured methods of analysis. Very clearly this practical application of machine learning will be on interest to the readership. Overall, the manuscript is well written though there are areas where the manuscript has potential to better serve the reader.
[Lines 39-43] While the term Privileged Learning in ML (LUPI in some disciples) has been around for a over a dozen years, it is still fairly new and if the introduction commented just a bit more on the reasoning for this article, it ends up with a introduction that may not be as puzzling to some readers. To place in the context of novel notions and uses of data would be informative for the improvement of the sepsis diagnosis.
[Lines 67-69] With Figure 1 immediately following the description of Equation (1), the reader has a disjointed sense of the narrative, sine this causes the reader to begin to understand the implications of Figure 1 in advance of Line 70 – 71. Wile the references do, indeed, provide adequate additional reference information, Figure 1 is not clearly described and the significance of the baseline t0 and privileged space t6 are lost. The manuscript refers to PI refers to a future section 2.3.3. so while not incorrect, the reader is jumping around trying of follow the logic LUPI in this context. Too cryptic to be of value, so you just skip over it.
[Lines 100-101] please expand on the Lagrangian term and the role the dual form the new equation plays in advancing from Eqn(3) to Eqn(4). This is a significant step that seems to come out in Lines 103-104.
[Lines128-129] It would seem that reciting the selection criteria for the cohort would be of interest, rather than asking the reader to dig into Reference 5.
[Lines 146-147] Was there any a-priori expectation the more generalized results for Cohort 3 would be more realistic in terms of accounting for Social Determinates of Health (SDOH) or any other reason. Just did not have enough data?
[Lines 204-205] Share with the reader the 4 sets of look-back – central to the PI strategy?
[Lines 206-226] Excellent discussion of results.
Discussion section. Informative summary description of results, but insight on “lessons learned” seems quite vague. For all the work performed, the conclusion seems to trail off and leave the reader wondering how informative the effort is from the LUPI and if the strategy for sepsis improvement was beneficial.
Comments on the Quality of English LanguageVery cryptic style that asks the readers to jump around and piece the story together.
Author Response
- Noted, the introduction has been updated to reflect the context of novel data usage in sepsis prognosis.
- The figure has been moved, and we hope that references to it now included in the text have improved the narrative and made it less cryptic.
- Expanded discussion of the Lagrangian term was originally excluded so as to not duplicate the mathematics of the Li et al. paper, but more information has been added to the Methods section.
- The selection criteria is briefly summarized in the section 2.2, and the full list of ICD codes used for the inclusion/exclusion criteria has now been added as an Appendix.
- The decision to place a patient on an arterial line is one that usually comes later in treatment, and once the patient is actively showing signs of being ill; the requirement of ABP necessarily excludes patients earlier in treatment who only have the less invasive ECG data available. Observing the demographics of the patients in the full retrospective dataset compared to the datasets that require telemetry data (ECG, ABP, etc.), while age and gender are fairly stable, there is a noted loss of racial/ethnic diversity, and as such, SDOH likely do play a role, but this has not been an active study of this group. Similar to reference 13 (Hernandez et al), the goal was to move outwards from more homogeneous to more heterogeneous datasets.
- The lists of features contained in the regular and privileged spaces have been added to tables (Table 1, page 8, Table 2, page 9) to make this clearer. The PI our group looked at was all contained to one 10-minute period, and did not include lookback periods so as to not overlap with the regular space.
- The discussion section has been modified. While this particular setup did not show LUPI to be helpful, the authors are hesitant to dismiss it outright, and are instead interested in exploring other potential use cases for LUPI in the context of sepsis prognosis.